# Faithful Temporal Question Answering over Heterogeneous Sources

## ABSTRACT

Temporal question answering (QA) aims to return crisp answers to user questions that involve temporal constraints, with phrases such as *"...in 2019"* or *"...before COVID"*. In the former, time is an *explicit* condition, in the latter it is *implicit*. State-of-the-art methods have limitations along three dimensions. First, when relying on neural inference, time constraints are often merely soft-matched, giving room to invalid or inexplicable answers. Second, questions with implicit time are poorly supported. Third, most systems tap into one source of information only, either a knowledge base (KB) or a text corpus. We propose a temporal QA system that addresses these shortcomings. First, it explicitly identifies and enforces temporal constraints for *faithful answering* with tangible evidence. Second, it includes techniques for properly handling *implicit questions*. Third, it operates over *heterogeneous sources*, covering KB, text and web tables in a unified manner. The method has three stages: (i) understanding the question and its temporal conditions, (ii) retrieving evidence from all sources, consistent with the temporal constraints, and (iii) faithfully answering the question from these pieces of evidence. As implicit questions are sparse in prior benchmarks, we introduce a principled method for generating diverse questions of this kind from heterogeneous sources. Experiments show superior performance over a suite of baselines.

## KEYWORDS

Question Answering, Temporal Questions, Explainability

## 1 INTRODUCTION

**Motivation**. Question answering (QA) comprises a spectrum of settings for satisfying users' information needs, ideally giving crisp, entity-level answers to natural-language utterances [44]. Temporal QA specifically focuses on questions with temporal conditions (e.g., [23, 30, 46]). Such questions pose challenges that are not properly met by universal QA systems. Consider the following example:

$q_1$: *Record company of Queen in 1975?*

The band Queen had different record companies over the years, so it is decisive to consider the *explicit temporal constraint* (*"in 1975"*). Other questions with explicit time are lookups of dates, such as:

$q_2$: *When was Bohemian Rhapsody recorded?*

*Conference'17, July 2017, Washington, DC, USA*
© 2023 Association for Computing Machinery.
ACM ISBN 978-x-xxxx-xxxx-x/YY/MM...$15.00
https://doi.org/10.1145/nnnnnnn.nnnnnnn

Another – underexplored and most challenging – situation is when questions involve *implicit temporal constraints*. These can involve the need to compare different time points or intervals, even when the user input does not explicitly state it. Examples are:

$q_3$: *Queen's record company when recording Bohemian Rhapsody?*
$q_4$: *Queen's lead singer after Freddie Mercury?*

For $q_4$, the system has to find out when Mercury died or left the band, in order to compute the correct answer that Brian May (the band's guitarist) took over as lead singer.

The research literature on temporal QA is substantial, including [9, 10, 15, 22–24, 30, 46, 55]. Most methods address all kinds of temporal questions, but are typically less geared for implicit questions. Some methods operate over curated knowledge bases (KBs) (e.g., [15, 22, 23]), while others are designed for processing text corpora such as news collections or Wikipedia full-text (e.g., [9, 34]).

**State-of-the-art limitations**. We observe three major issues:

(i) Many methods solely use "soft-matching" techniques, based on latent embeddings or language models for computing answers. This may lead to invalid answers, where the non-temporal part of a question is matched, but the temporal constraint is violated. For example, a question about *"Queen's record company in 1990?"* may erroneously return EMI instead of the correct value Parlophone, simply because EMI is much more prominent and was Queen's company on most albums. Even when the output itself is correct, this could result from the prominence of the answer alone. For example, *"Who was Queen's lead singer in 1975?"* could return the most popular Freddie Mercury without checking the time part at all. When we vary the question into *"...in 2000?"*, many systems would still yield Freddie Mercury, although he was dead then. These are indicators that the system has incomplete inference and is unable to explain the answer derivation. We call this phenomenon *unfaithful QA*.

ii) A weak spot of temporal QA systems is the handling of *implicit questions*. These are infrequent in established benchmarks. Some methods [15, 22, 33] aim to transform the implicit conditions into explicit temporal constraints, based on classifying phrases starting with "during", "before" etc. However, they heavily rely on hand-crafted rules which are rather limited in scope and cannot robustly handle unforeseen utterances.

(iii) Existing methods operate only over a *single information source*, typically either a KB or a text corpus. This implies limited coverage: KBs such as Wikidata are inherently incomplete and often lack refined detail about events, whereas text collections such as Wikipedia full-text are harder to extract answers from and are often bound to fail on complex questions [11, 15]. QA over heterogeneous sources, including also web tables, has been recently addressed by [12, 36], but these methods are designed for broad questions without consideration of temporal conditions.

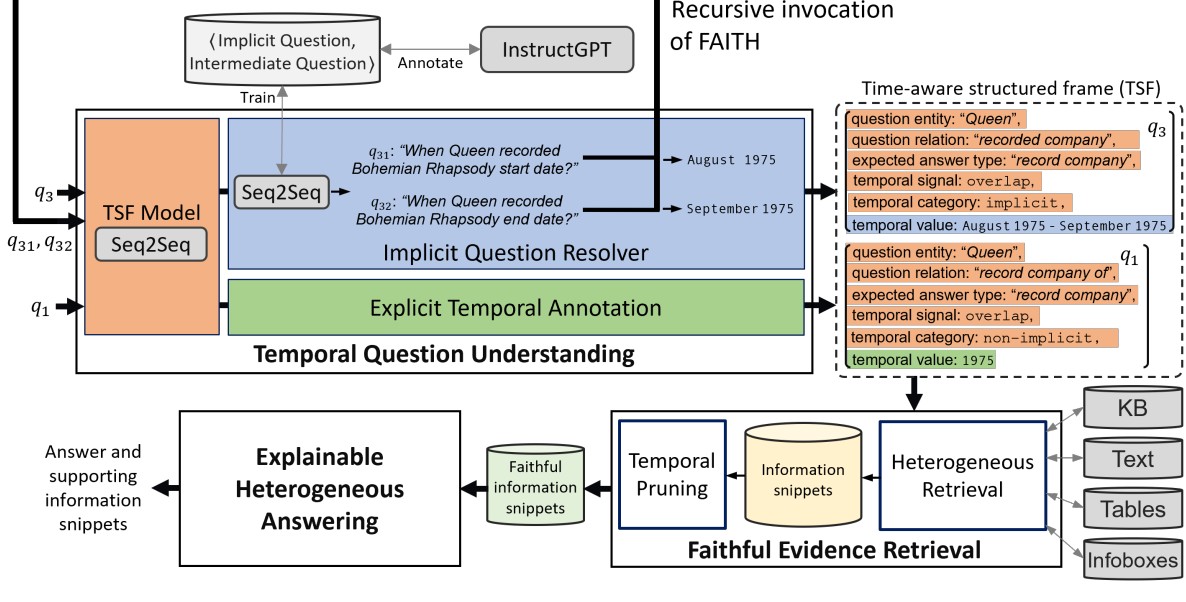

**Figure 1: Overview of the FAITH. The figure illustrates the process for answering $q_3$ ("Queen's record company when recording Bohemian Rhapsody?") and $q_1$ ("Record company of Queen in 1975?"). For answering $q_3$, two intermediate questions $q_{31}$ and $q_{32}$ are generated, and run recursively through the entire temporal QA system.**

**Approach**. To overcome these limitations, we propose FAITH (FAIthful Temporal question answering over Heterogeneous sources), a temporal QA system that operates over *heterogeneous* sources, seamlessly combining a KB, a text corpus and web tables. Inspired by the architecture of [12] for non-temporal QA, FAITH consists of three main stages:

 (i) **temporal question understanding** for representing the question intent into a structured frame, with specific consideration of the temporal aspects;
 (ii) **faithful evidence retrieval** for identifying relevant pieces of evidence from KB, text and tables, with time-aware filtering to match the temporal conditions;
 (iii) **explainable heterogeneous answering** to compute entity-level answers and supporting evidence for explanation.

A key novelty in the question understanding is that implicit constraints are resolved into explicit temporal values by generating intermediate questions and recursively calling FAITH itself. For example, the implicit condition *"when recording Bohemian Rhapsody"* in $q_3$ is transformed into *"when Queen recorded Bohemian Rhapsody?"*, and the recursive invocation of FAITH returns the explicit condition August 1975 – September 1975. This derived explicit condition is then used in a similar vein as the explicit condition 1975 in $q_1$, making it easier to answer the information need. Note that this is not just question rewriting, but is driven by the full-fledged QA system itself over the full suite of heterogeneous sources.

A second key novelty is that, in contrast to most prior works including large language models, FAITH provides *tangible provenance* for the answer derivation. By providing users with explanatory evidence for answers, FAITH is a truly faithful temporal QA system.

Existing benchmarks for temporal QA focus on a single information source at hand (either a KB or a text corpus), and include only few questions with implicit constraints (so the weak performance on these hardly affects the overall benchmark results). Therefore, we devise a new method for automatically creating temporal questions with *implicit constraints*, with systematic controllability of different aspects, including the relative importance of different source types (text, infoboxes, KB), coverage of topical domains (sports, politics etc.), fractions of prominent vs. long-tail entities, question complexity, and more. This way, we construct a new data resource named TIQ with 10,000 questions and ground-truth answers accompanied by supporting evidence.

**Contributions**. The salient contributions of this work are:
 • the first temporal QA system that operates over heterogeneous information sources, and can provide faithful answers with explanatory evidence;
 • a mechanism that transforms implicit temporal constraints into explicit conditions, by recursively calling the QA system itself;
 • a principled method for automatic construction of diverse and difficult temporal questions, releasing a new benchmark.

## 2 FAITH METHOD

Figure 1 provides an overview of the system architecture, illustrated with the processing of the running examples $q_3$ and $q_1$. The following subsections present the three main components (understanding, retrieval, answering), and will refer to these examples.

### 2.1 Temporal Question Understanding

The goal of the first stage is to capture the temporal information need in a frame-like structure. Notably, this stage identifies and

categorizes temporal constraints in the user input, which is later used for pruning temporally-inconsistent answer candidates.

**TSF.** Inspired by [12] and [19] (both addressing other, non-temporal, kinds of QA), we propose to learn a *time-aware structured frame (TSF)* for an incoming temporal question. This representation includes both general-QA-relevant slots, like

- *question entity*,
- *question relation*,
- *expected answer type*,

and temporal-QA-relevant slots:

- *temporal signal*, indicating the kind of temporal relation,
- *temporal category*, indicating the type of temporal constraint,
- *temporal value*, indicating the time point or interval of interest (if inferrable).

The question entity and relation are taken from the surface form of the question (i.e. *not* linked to KB) to allow for uniform treatment of heterogeneous sources. The expected answer type is learned from the training data, in which the KB-type of the gold answer is used.

The *temporal signal* can be `overlap` (e.g., from cues like *"in"*, *"during"*), `before` (e.g., from cues like *"before"*, *"prior to"*), or `after` (e.g., from cues like *"after"*, *"follows"*). The *temporal value* can be a *year*, *date* or *time period*. Both, the temporal signal and value, are derived by identifying and normalizing key phrases in the input question. For example, the TSF for $q_1$ is:

⟨ question entity: *"Queen"*,
question relation: *"record company of"*,
expected answer type: *"record company"*,
temporal signal: `overlap`,
temporal category: `non-implicit`,
temporal value: `1975` ⟩

Note that in case the question does not specify temporal constraints (e.g. $q_2$), the respective fields are simply kept empty.

**Resolving implicit questions.** For the challenging case of implicit questions, such as $q_3$ or $q_4$, the temporal value cannot be extracted from the question directly. To resolve this problem, we devise a novel mechanism, the *implicit question resolver*, based on recursively invoking the temporal QA system itself. To this end, the temporal constraint in the question is identified and transformed into an *intermediate question*. For instance, the intermediate question for $q_4$ would be *"when Freddie Mercury lead singer of Queen?"*. For $q_3$, the temporal value should be a time interval (`August 1975 – September 1975`). Thus, two intermediate questions are required: (i) $q_{31}$: *"When Queen recorded Bohemian Rhapsody start date?"*, and (ii) $q_{32}$: *"When Queen recorded Bohemian Rhapsody end date?"*. Although these formulations are ungrammatical, the QA system can process them properly, being robust to such inputs.

The intermediate questions are fed into FAITH as a recursive call, to obtain the explicit temporal value for filling the TSF of the original question. The TSF for $q_3$ thus becomes:

⟨ question entity: *"Queen"*,
question relation: *"recorded company"*,
expected answer type: *"record company"*,
temporal signal: `overlap`,

temporal category: `implicit`,
temporal value: `August 1975 – September 1975` ⟩

Note the similarity to the TSF of the explicit temporal question $q_1$.

The intermediate questions are generated by a fine-tuned sequence-to-sequence (Seq2seq) model, specifically BART [26]. A major obstacle, though, is that no prior dataset has suitable annotations and collecting such data at scale is prohibitive. Therefore, we generated training data using InstructGPT [37], leveraging its *in-context learning* [3] capabilities. We randomly select 8 implicit questions from our train set and label them manually. For each question, we give the intermediate question and the expected answer type as output. The exact prompts used are shown in Table 7 and 8 in the Appendix. The expected answer type of an intermediate question can be `date` or `time interval`. When the expected answer type is a time interval (e.g. for $q_3$), two intermediate questions are created, appending *"start date"* and *"end date"* to the generated intermediate question, respectively (see $q_{31}$ and $q_{32}$ as example).

The prompts are used to annotate all implicit questions in the train and dev sets, obtaining large-scale data for fine-tuning the BART model. Note that GPT is used only for the generation of training data for fine-tuning. It is not used by FAITH at run-time to avoid its (computational, monetary and environmental) costs and dependency on black-box models.

**Constructing the TSF.** We also use a fine-tuned Seq2seq model, again BART, for generating the values for the question entity, question relation, expected answer type, temporal signal, and temporal category slots of the TSF representation.

The training data for fine-tuning the TSF construction model is obtained via (i) distant supervision (for question entity and question relation), (ii) KB-type look-ups (for expected answer type), and (iii) annotations in the benchmark (for temporal signal and temporal category). Further detail is given in the Appendix C.

The temporal values are obtained via the recursive mechanism discussed above for implicit questions, and via SUTime [6] and regular expression matching for explicit questions. Phrases like *"today"* or *"current"* are considered as well and properly normalized. We use the creation time of the question [5], as provided in the benchmarks, as reference time.

The TSF generated in this understanding stage is used for representing the temporal information need in the subsequent retrieval and answering stages.

## 2.2 Faithful Evidence Retrieval

This stage retrieves relevant evidence matching the temporal constraint, that is expressed by the temporal signal and temporal value in the TSF. We perform two steps: (i) heterogeneous retrieval, and (ii) temporal pruning.

**Heterogeneous retrieval.** This largely follows the general-purpose QA method of [12], using an entity-centric retriever, CLOCQ [11], to identify and link entity mentions in the input. The input here is the concatenation of the *question entity*, the *question relation*, and the *expected answer type* of the TSF. For the linked entities, we retrieve the Wikipedia pages for extracting text, tables, and infoboxes. Further, the connected KB-facts are obtained from Wikidata.

All retrieved pieces of evidence are *verbalized* [36] into textual sentences, for uniform treatment. The KB-facts are verbalized by

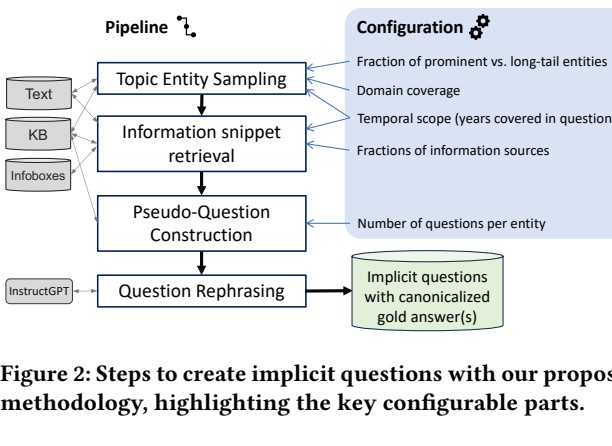

**Figure 2: Steps to create implicit questions with our proposed methodology, highlighting the key configurable parts.**

concatenating their individual parts; the text evidence is split into sentences; table rows are transformed by concatenating the individual ⟨column headers, cell value⟩ pairs; infoboxes are handled by linearizing all attribute-value pairs.

**Temporal pruning**. Explicit temporal expressions in the retrieved pieces of evidence are identified and normalized similarly as in the understanding stage. Evidence that does not match the temporal criteria is pruned out. We address two kinds of situations:

(i) the question aims for a temporal value as answer and does not have any temporal constraints (e.g., "When … ?");

(ii) the question has a temporal constraint which needs to be matched by the evidence.

In the first case, all evidence that does not contain any temporal values is dropped. In the second case, we remove pieces of evidence that do not match the temporal constraint.

The retrieval output is a smaller set of evidence pieces, faithfully reflecting the temporal constraints of the question. The final answer and its explanatory evidence are computed from this pool.

## 2.3 Explainable Heterogeneous Answering

In the final stage, the answer is derived from this set of evidence snippets that is already known to satisfy the temporal conditions.

Since this part is not the main focus of this work, we employ a state-of-the-art answering model for general-purpose QA. We use the answering stage of EXPLAIGNN [12] that is based on graph neural networks (GNNs), and provides supporting evidence for the predicted answer. Thus, we ensure that the answer can be traced back through the entire system including the answering stage, for end user explainability. The input query to the GNNs, is the concatenation of the question entity, question relation, and expected answer type.

## 3 TIQ BENCHMARK

Benchmarks for temporal QA, like TempQuestions [21], TimeQuestions [23] or TempQA-WD [33], have only few implicit questions, falling short to evaluate one of the key challenges in temporal QA. CronQuestions [46] has a larger fraction of implicit questions, but these are based on a small set of hand-crafted rules. Therefore, the questions lack *syntactic diversity*. Further, questions in these benchmarks are always answerable using a single information source (either KB or text corpus).

Therefore, we construct a new benchmark with focus on implicit temporal questions. The obvious idea of using crowdsourcing is expensive and error-prone. Also, crowdworkers increasingly use LLMs as a shortcut [51]. Thus, we pursue an automated process instead. Questions and ground-truth answers are generated from multiple sources: Wikipedia text and infoboxes, and the Wikidata KB. The resulting resource and all code will be released upon publication of the paper.

## 3.1 Construction Methodology

**Overview**. An implicit question has two parts: the *main question* that specifies the information need disregarding time (e.g. *"Queen's lead singer"* for $q_4$), and the *implicit part* that provides the temporal constraint (e.g. *"after Freddie Mercury"* for $q_4$). The key idea is to build each of the two parts from independent pieces of evidence, denoted as *information snippets*. The two snippets can come from very different sources, but need to be thematically related. This construction process operates as follows:

(i) sample a set of topic entities to start with;

(ii) retrieve temporal information snippets for each such topic entity from Wikipedia text, Wikipedia infoboxes, and Wikidata;

(iii) concatenate information snippets using a suitable temporal signal and construct an interrogative sentence, a *pseudo-question*;

(iv) rephrase the pseudo-question into a natural question using a generative model.

An overview of this process is provided in Figure 2. Naturally, implicit constraints are global events (e.g. the CoVid pandemics), or major events for a specific entity (e.g. a prestigious award).

**Sampling topic entities**. To obtain significant events, we use the following proxy: we start with Gregorian calendar year pages in Wikipedia (e.g. https://en.wikipedia.org/wiki/2023) that list notable events per year (e.g. 1880 - 2025). From these year pages, we collect information snippets about notable events. The entities in snippets constitute the set of topic entities (href anchors are used for entity linking [16]).

**Retrieving the grounding information snippets**. The snippets collected from the year pages are further augmented by snippets from the first five sentences (≃ first passage) of the topic entity's Wikipedia article, the respective Wikipedia infobox, and the entity's facts from Wikidata.

As candidates for the main question part, we consider all information snippets that are retrieved for a topic entity from Wikipedia text, infoboxes and Wikidata, irrespective of their salience. To avoid questions that are trivially answerable without considering the temporal condition, multiple candidate snippets are retrieved for the main question, with different temporal scopes (e.g., a band's singers from different epochs). This is implemented by measuring semantic similarity among candidates using SBERT [43].

**Creating a pseudo-question**. Among the retrieved snippets for an entity, we identify pairs of candidate snippets that can be connected by a temporal conjunction/preposition (*"during"*, *"after"* and *"before"*). For such a pair, the temporal scopes have to be consistent with the temporal conjunction. A valid pair for the conjunction

"*during*" would be: "*Alicia Keys followed up her debut with The Diary of Alicia Keys, which was released in December 2003.*" (main question part from Wikipedia text) and "*Norah Jones, award received, Grammy Award for Best New Artist, follows, Alicia Keys, point in time, 2003.*" (implicit part from KB).

A *pseudo-question* is created by concatenating the main part with the conjunction and the implicit part. The answer is the entity from the main part (`The Diary of Alicia Keys`). The answer is substituted by the prefix "*what*" followed by the most frequent KB-type of the answer (`album` in this case).

The pseudo-question for the example is: "*What album Alicia Keys followed up her debut with which was released, during, Norah Jones award received Grammy Award for Best New Artist follows Alicia Keys?*", which is an ungrammatical and unnatural formulation.

**Rephrasing to a natural question**. Therefore, in the last step, we rephrase the pseudo-question to a natural formulation. We use InstructGPT [37] with 8 demonstration examples (pseudo-questions and their natural re-phrasings), to generate the final question. We experimented with different prompts; the best one is shown in Table 9 in the Appendix.

The example pseudo-question is rephrased to: "*What album did Alicia Keys release when Norah Jones won the Grammy Award for Best New Artist?*"

## 3.2 Benchmark Characteristics

**Topic entities**. For creating TIQ (Temporal Implicit Questions) we started with the years `1801-2025` and obtained an initial set of 229,318 entities. From this set, we uniformly sampled 10,000 topic entities based on their frequency, to capture a similar amount of long-tail and more prominent entities (see Table 1 for details). These fractions can be configured as required. Since some entity types were over-represented in the calendar year pages (e.g. politicians or countries), we also ensured that individual entity types are not taking up more than 10% of the topic entities. In general, the topic entity set allows to control the domain coverage within the generated implicit questions, by choosing entities of the desired types.

We did not specifically configure the proportions to which the individual information sources are used within the questions, since we observed a naturally diverse distribution. Figure 3 shows the distribution among source combinations for initiating the main and implicit part. The questions are finally split into train (6,000), dev (2,000), and test sets (2,000). Table 1 shows the basic statistics and Table 2 shows representative questions of the TIQ benchmark.

**Meta-data**. TIQ provides implicit questions and gold answers, as strings as well as canonicalized to Wikipedia and Wikidata. The meta-data includes the information snippets grounding the question, the sources these were obtained from, the explicit temporal value expressed by the implicit constraint, the topic entity, the question entities detected in the snippets, and the temporal signal.

## 4 EXPERIMENTS

## 4.1 Experimental setup

**Benchmarks**. We conduct experiments on our new TIQ benchmark and TIMEQUESTIONS [23], which has been actively used in recent work on temporal QA. TIMEQUESTIONS also has ordinal questions (e.g. "*what was the first album by Queen?*"). For such questions, we

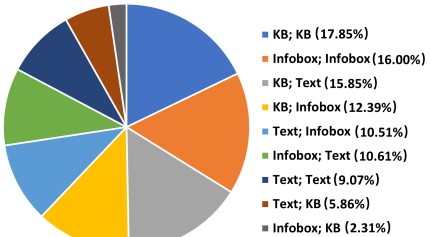

Figure 3: Distribution of questions over input source combinations (source for main part ; source for implicit part).

**Table 1: Basic statistics for TIQ.**

| Sources | Wikipedia text, infoboxes, and Wikidata |
|---|---|
| Questions | 10,000 (train: 6,000, dev: 2,000, test: 2,000) |
| Avg. question length | 17.96 words |
| Avg. no. of question entities | 2.45 |
| Unique topic entities covered | 10,000 |
| Long-tail topic entities covered | 2,542 (with < 20 KB-facts) |
| Prominent topic entities covered | 2,613 (with > 500 KB-facts) |

apply the same method as outlined in Section 2, without applying any temporal filtering.

**Metrics**. We use the standard QA metrics precision at 1 (P@1), mean reciprocal rank (MRR), and hit at 5 (Hit@5) [44]. Metrics are averaged over all questions.

**Baselines**. We compare FAITH with a suite of baselines, covering a diverse range of competitors:

- **Generative LLMs**. We compare with **INSTRUCTGPT** [37] ("text-davinci-003") and **GPT-4** [35] ("gpt-4") using the OpenAI API[1]. We tried different prompts, and found the following to perform best: "*Please answer the following question by providing the crisp answer entity, date, year, or number.*". For computing P@1, we check whether the generated answer string matches with the label or any alias of the gold answer. If this is the case, P@1 is 1, else 0. Other (ranking) metrics are not applicable for LLMs.
- **Heterogeneous QA methods**. Further, we compare with recent general-purpose QA systems operating over heterogeneous sources: **UNIQORN** [40], **UNIK-QA** [36], and the vanilla **EXPLAIGNN** [12].
- **Temporal QA methods**. We also compare with state-of-the-art methods for temporal QA: **TEMPOQR** (TempoQR-Hard) [30], **CRONKGQA** [46], and **EXAQT** [23].

Finally, we show results for a variant of our approach, which does *not prune out* evidence temporally-inconsistent with the temporal constraint, i.e. drops the temporal pruning component. We term this variant **UN-FAITH**.

**Configuration**. Wikidata [52] is used as the KB for FAITH and all baselines. We use Wikipedia text, tables and infoboxes as additional information sources for all methods operating over heterogeneous sources. The BART models are initialized and trained using Hugging Face[2]. EXPLAIGNN is run using the public code[3].

In FAITH, we choose the candidate at rank 1 as the answer for intermediate questions in the implicit question resolver. In case too

---

[1]https://platform.openai.com
[2]https://huggingface.co
[3]https://github.com/PhilippChr/EXPLAIGNN

**Table 2: Representative questions from the TIQ benchmark. The sources below indicate the source that was used for populating the [main question part; implicit question part] of the implicit question.**

| 1. *Who bought the Gainesville Sun after it was owned by Cowles Media Company?* | 2. *During Colin Harvey's senior football career, which club was he a member of while he played for the England national football team?* | 3. *Which album released by Chris Brown topped the Billboard 200 when he was performing in Sydney?* | 4. *What television series was Hulk Hogan starring in when he signed with World Championship Wrestling?* | 5. *Who was Bristol Palin's partner before she participated in the fall season of Dancing with the Stars, and reached the finals, finishing in third place?* |
|---|---|---|---|---|
| `The New York Times Company` [Text; KB] | `Everton F.C.` [Infobox; KB] | `Fortune` [Text; Infobox] | `Thunder in Paradise` [Text; Text] | `Levi Johnston` [Infobox; Text] |
| 6. *During the onset of the COVID-19 pandemic, who was the New York City head of government?* | 7. *Who was the chief executive officer at Robert Bosch GmbH before revenue reached € 78.74 billion?* | 8. *After graduating from the Rostov-on-Don College of Economics and Finance, which political party did Gyula Horn join?* | 9. *Which national football team did Carlos Alberto Torres manage before joining Flamengo?* | 10. *What university did Robert Lee Moore work for after Northwestern University?* |
| `Bill de Blasio` [KB; Text] | `Volkmar Denner` [KB; Infobox] | `Hungarian Working People's Party` [Infobox; Text] | `Oman national football team` [Infobox; Infobox] | `University of Pennsylvania` [KB; KB] |

**Table 3: Performance comparison of FAITH with baselines on the *test* sets of TIQ and TIMEQUESTIONS.**

| Benchmark → | TIQ | | | TIMEQUESTIONS | | |
|---|---|---|---|---|---|---|
| Method ↓ | P@1 | MRR | Hit@5 | P@1 | MRR | Hit@5 |
| INSTRUCTGPT [37] | 0.237 | n/a | n/a | 0.224 | n/a | n/a |
| GPT-4 [35] | 0.236 | n/a | n/a | 0.306 | n/a | n/a |
| UNIQORN [40] | n/a | n/a | n/a | 0.331 | 0.409 | 0.538 |
| UNIK-QA [36] | 0.425 | 0.480 | 0.540 | 0.424 | 0.453 | 0.486 |
| EXPLAIGNN [12] | 0.446 | 0.584 | 0.765 | 0.525 | 0.587 | 0.673 |
| TEMPOQR [30] | 0.011 | 0.018 | 0.022 | 0.438 | 0.465 | 0.488 |
| CRONKGQA [46] | 0.006 | 0.011 | 0.014 | 0.395 | 0.423 | 0.450 |
| EXAQT [23] | 0.232 | 0.378 | 0.587 | 0.565 | 0.599 | 0.664 |
| FAITH (Proposed) | 0.462 | 0.582 | 0.749 | 0.530 | 0.578 | 0.644 |
| Un-FAITH | 0.480 | 0.627 | 0.827 | 0.596 | 0.656 | 0.730 |

many evidences are obtained as input to the answering stage, we apply SBERT[43] to score evidences and retain only the top-100 evidences. Further detail is given in the Appendix F. We follow an epoch-wise evaluation strategy for each module and baseline, and take the version with the best performance on the respective dev set. All training processes and experiments are run on a single GPU (NVIDIA Quadro RTX 8000, 48 GB GDDR6).

## 4.2 Main results

Answering performance of FAITH and baselines on TIMEQUESTIONS and on TIQ are in Table 3.

**FAITH outperforms baselines on TIQ**. The main insight from Table 3 is that FAITH surpasses all baselines on the TIQ dataset for P@1, which is the most relevant metric, demonstrating the benefits of our proposed method for answering implicit temporal questions. Temporal QA methods operating over KBs lack the required coverage on the TIQ dataset, and perform worse than general-purpose QA methods operating over heterogeneous sources. The general-purpose QA system EXPLAIGNN comes close to the performance of FAITH, and even slightly improves on the MRR and Hit@5 metrics. Note, however, that EXPLAIGNN and all other baselines do not verify that temporal constraints are met during answering. Thus, the most prominent among answer candidates may simply be picked up, even if no temporal information is provided or matching. Such possibly "accidental" (unfaithful) answers are, by design, not considered by FAITH.

**Trade-off between faithfulness and answering performance**. Results for Un-FAITH illustrate the effect of this phenomenon on our approach: especially the MRR and Hit@5 results are substantially improved. Consequently, Un-FAITH outperforms all competitors on both benchmarks. However, its answers are not always faithfully grounded in evidence sources. These results emphasize the trade-off between faithfulness and answering performance.

**FAITH shows robust performance on TIMEQUESTIONS**. FAITH also shows strong performance on the TIMEQUESTIONS benchmark, on which it outperforms all baselines on P@1, except for EXAQT. This indicates the robustness of FAITH across different datasets. Existing methods for temporal QA show major performance gaps between the two benchmarks: the P@1 of the strongest method on TIMEQUESTIONS, EXAQT, substantially drops from 0.565 at P@1 to 0.232 on the TIQ benchmark. Note that all methods are trained on the specific benchmark, if applicable.

**LLMs fall short on temporal questions**. Another key insight from Table 3 is that current LLMs are clearly not capable of answering temporal questions. INSTRUCTGPT and GPT-4 can merely answer ≃ 23-30% of the questions correctly, and are constantly underperforming FAITH and baselines operating over heterogeneous sources. One explanation is that reasoning with continuous variables, such as time, is a well-known weakness of LLMs [14].

## 4.3 In-depth analysis

**FAITH refrains to answer in absence of consistent evidence**. If there is no temporal information associated with the evidence of candidate answers, or the temporal information does not satisfy the temporal constraint, FAITH will refuse answering the question. For example, for the question *"Who did Lady Jane Grey marry on the 25th of May 1533?"*, there is no answer satisfying the temporal constraint because *Lady Jane Grey* did not marry anyone *on the 25th of May 1533*, since she was only born two years later in 1955. However, all of the baselines provide an answer to the question, without indicating that the temporal constraint is violated.

Since such questions without a temporally-consistent answer are not available at large scale, we randomly sample 500 explicit questions from TIMEQUESTIONS, and replace the temporal value with a random date (e.g. `12 October 6267`). None of the resulting questions has a temporally-consistent answer. As expected, the competitors still provide a ranked list of answers[4]. In contrast,

---

[4]Except for the LLMs for which we are not able to investigate the behavior at scale, since they would often generate longer texts.

**Table 4: Ablation study using different source combinations as input for FAITH on *dev* sets. Note that FAITH is trained using *all sources* as input for all cases.**

| Benchmark → | TIQ | | | TIMEQUESTIONS | | |
|---|---|---|---|---|---|---|
| Method ↓ | P@1 | MRR | Hit@5 | P@1 | MRR | Hit@5 |
| **KB** | 0.289 | 0.360 | 0.457 | 0.424 | 0.460 | 0.503 |
| **Text** | 0.170 | 0.236 | 0.320 | 0.218 | 0.272 | 0.340 |
| **Infoboxes** | 0.169 | 0.223 | 0.296 | 0.119 | 0.143 | 0.172 |
| **Tables** | 0.032 | 0.050 | 0.076 | 0.078 | 0.097 | 0.120 |
| **KB+Text** | 0.386 | 0.492 | 0.627 | 0.526 | 0.574 | 0.636 |
| **KB+Tables** | 0.294 | 0.372 | 0.475 | 0.446 | 0.486 | 0.537 |
| **KB+Infoboxes** | 0.302 | 0.475 | 0.604 | 0.455 | 0.494 | 0.542 |
| **Text+Tables** | 0.177 | 0.244 | 0.325 | 0.248 | 0.304 | 0.374 |
| **Text+Infoboxes** | 0.276 | 0.357 | 0.456 | 0.261 | 0.316 | 0.389 |
| **Tables+Infoboxes** | 0.172 | 0.230 | 0.302 | 0.166 | 0.200 | 0.239 |
| **All sources** | 0.469 | 0.584 | 0.736 | 0.543 | 0.591 | 0.653 |

**Table 5: Ablation studies of FAITH on dev sets.**

| Benchmark → | TIQ | TIMEQUESTIONS |
|---|---|---|
| Method ↓ | P@1 | P@1 |
| **FAITH** | 0.469 | 0.543 |
| **w/o temporal pruning** | 0.475 | 0.604 |
| **w/o implicit question resolver** | 0.427 | 0.564 |
| **w/o GNN-based answering** | 0.304 | 0.394 |

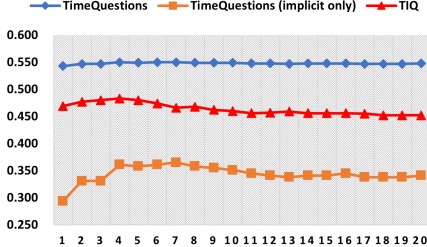

**Figure 4: P@1 of FAITH when considering top-$k$ answers for the intermediate question(s).**

FAITH successfully refrained from answering for 467 of the 500 questions (93.4%). Upon investigating the failure cases, we noticed that the date recognition identifies four-digit numbers as years matching with the constraint (e.g. in *"Veysonnaz, Population (2018-12-31), SFOS number, 6267"*).

**Fallback to Un-FAITH**. Completely refraining from answering could also be sub-optimal: the user might have made a typo (e.g. *"May 1533"* instead of *"May 1553"*). We investigated to fall back to Un-FAITH in such scenarios, which could be indicated to end users with an appropriate warning. Performance on TIMEQUESTIONS was slightly improved (P@1 from 0.530 to 0.536), while on TIQ FAITH always found some matching evidence (i.e. no changes/improvement). We further investigated to fall back to Un-FAITH in case FAITH answered incorrectly. This substantially improved performance on both datasets: the P@1 metric increased from 0.462 to 0.614 on TIQ and from 0.530 to 0.667 on TIMEQUESTIONS.

**Integrating heterogeneous sources is decisive**. We further investigated the effect of integrating heterogeneous sources into FAITH, and tested giving each individual source independently, and their pairwise combinations as input, in comparison to the default

setting with "All sources". Results are in Table 4. Each information source contributes to the performance of FAITH, and integrating more information sources consistently enhances all metrics.

**Ablation studies**. We tested variations of our pipeline on the dev sets. Table 5 shows results for Un-FAITH (w/o temporal pruning), results without the implicit time resolver, and results with a Seq2seq model for answering (we used BART) instead of the GNN-based approach. Using a GNN-based answering approach plays a crucial role, and enhances not only answering performance, but also explainability. The implicit question resolver is decisive on TIQ, but slightly decreases performance on TIMEQUESTIONS. Un-FAITH also shows strong performance on the dev sets. However, all modules contribute to the explainability and faithfulness of our approach.

**Relaxed temporal pruning**. In FAITH we consider the top-1 answer(s) as temporal value(s) within the implicit question resolver. Since there may be errors in predicting these answers (the P@1 on TIQ is 0.532, further analysis in Appendix G.3), we investigated considering top-$k$ answers for the intermediate questions. Figure 4 shows results of this analysis, varying $k$ from 1 to 20. We observed that the P@1 improves as $k$ is increased initially. A maxima is reached for a value of 4 or 5. As $k$ increases, the set of candidate snippets converges resulting in a stable P@1.

**Anecdotal examples**. Table 6 shows sample cases for which FAITH provided the correct answer, and illustrates the answer derivation process providing traceable evidence for end users.

**Error analysis**. To better understand failure cases, we conducted an error analysis measuring the *answer presence* (i.e. whether the gold answer is among answer candidates) throughout the pipeline. We identified the following error cases (% of *failure cases* in TIQ/ TIMEQUESTIONS): (i) the answer was not found in the initial retrieval stage (2.6/28.9), (ii) the answer is lost during temporal pruning (21.2/26.8), (iii) the answer is lost during scoring/graph shrinking (8.6/10.2), (iv) the answer is not considered among top-5 answers (14.3/9.8), (v) the answer is among top candidates but not at rank 1 (53.3/24.3). Note that these numbers add up to 100% for both benchmarks, respectively (100% of failure cases).

The temporal pruning and fine-grained answer ranking are the most error-prone steps on both benchmarks, leading to 88.8% (TIQ) and 60.9% (TIMEQUESTIONS) *of the failure cases*. Improving the question understanding and retrieval could help with the first case. Enhancing the answering stage with mechanisms specific for temporal QA could help mitigate the second failure case.

## 5 RELATED WORK

**General-purpose QA**. Question answering is a long-standing research topic with extensive work using single sources like KBs (e.g. [2, 59, 61]) or text (e.g. [7, 20, 42]) for deriving answers.

It has been shown in multiple works that integrating different information sources can substantially improve performance of general-purpose QA [8, 17, 45, 49, 50, 57, 58]. More recently, UNIK-QA [36] proposed to verbalize information snippets from a KB, text, tables and infoboxes for integrating such heterogeneous sources. These uniformed text snippets are then given as input to a Fusion-in-decoder (FiD) model [20] for generating the answer. In UDT-QA [28] the verbalization was improved. EXPLAIGNN [12] also verbalizes information pieces, but then constructs a graph considering their relationship based on shared entities. On this graph,

**Table 6: Anecdotal examples that Faith answered correctly in Tɪϙ and TɪᴍᴇϘᴜᴇsᴛɪᴏɴs. Evidence shows the supporting information snippets along with their source provided in brackets. The part mentioning the predicted answer is in bold, and the detected temporal values are underlined. For the first example from the Tɪϙ benchmark, we show the answering process of the intermediate question, which can be used by end users to verify the entire answer derivation of the system.**

| Benchmark | Tɪϙ |
| --- | --- |
| Question | *After managing FC Nantes, which football club did Antoine Raab take on next?* |
| Answer | **Stade Lavallois** |
| TSF | ⟨ question entity: *"Antoine Raab, FC Nantes football"*, question relation: *"After managing which club did take on next"*, expected answer type: *"association football club"*, temp. signal: after, temp. category: implicit, temp. value: [1946, 1949] ⟩ |
| Evidence | *Antoine Raab, Managerial career, 1949–1950, **Stade Lavallois**.* (ꜰʀᴏᴍ Iɴꜰᴏʙᴏx) |
| Intermediate questions | (i) *When Antoine Raab managed FC Nantes start date?* 
 (ii) *When Antoine Raab managed FC Nantes end date?* |
| Answers (to int. questions) | (i) **1946**, (ii) **1949** |
| TSFs (for int. questions) | (i) ⟨ question entity: *"FC Nantes, start, Antoine Raab"*, question relation: *"When managed date"*, expected answer type: *"year"*, temp. signal: _; temp. category: non-implicit; temp. value: _ ⟩ 
 (ii) ⟨ question entity: *"FC Nantes, end, Antoine Raab"*, question relation: *"When managed date"*, expected answer type: *"year"*, temp. signal: _; temp. category: non-implicit; temp. value: _ ⟩ |
| Evidence (for int. questions) | (i, ii) *Antoine Raab, Managerial career, **1946–1949**, FC Nantes.* (ꜰʀᴏᴍ Iɴꜰᴏʙᴏx) 
 (ii) *Antoine Raab, After the liberation of Nantes in 1944 Raab joined FC Nantes and played for the club until **1949**.* (ꜰʀᴏᴍ Tᴇxᴛ) |
| Benchmark | TɪᴍᴇϘᴜᴇsᴛɪᴏɴs |
| Question | *What award did Thomas Keneally receive in the year 1982?* |
| Answer | **Booker Prize** |
| TSF | ⟨ question entity: *"Thomas Keneally"*, question relation: *"What award did receive in the year 1982"*, expected answer type: *"science award"*, temp. signal: overlap, temp. category: non-implicit, temp. value: 1982 ⟩ |
| Evidence | *Man **Booker Prize**, winner, Thomas Keneally, point in time, 1982, for work, Schindler's Ark.* (ꜰʀᴏᴍ KB) 
 *Thomas Keneally, Awards is **Booker Prize**, is Schindler's Ark, winner 1982.* (ꜰʀᴏᴍ ᴛᴀʙʟᴇ) 
 *Thomas Keneally, He is best known for his non-fiction novel Schindler's Ark, the story of Oskar Schindler's rescue of Jews during the Holocaust, which won the **Booker Prize** in 1982.* (ꜰʀᴏᴍ Tᴇxᴛ) |

iterative graph neural networks are used for predicting the answer. All of these methods are tied to general-purpose QA, and are not able to faithfully answer more complex temporal questions.

Another direction is to directly apply large language models (LLMs) for general-purpose QA, since LLMs are known to store world knowledge in their vast parameter space [3, 13, 39, 41]. However, LLMs cannot present traceable provenance for the provided outputs, and faithfulness and explainability are key concerns [1, 29, 32]. Further, time is a continuous variable, and thus LLMs often struggle to properly model the temporal dimension [14]. **Temporal QA**. Temporal questions pose challenges that are out-of-scope for general-purpose QA systems. Therefore, there has been extensive research that specifically targets temporal QA [9, 10, 15, 22–24, 27, 30, 33, 46–48, 54, 55, 60], which can largely be divided into work using the KBs for deriving the answer (e.g. [23, 30, 33]), and work using text (e.g. [9, 34]). *Methods operating over KBs*, include template-based [15, 22, 33], KB-embedding-based [10, 30, 46, 55], and graph-based methods [23, 48, 60]. *Methods using textual inputs* typically involve an extractive or generative reader [9, 34].

Exᴀϙᴛ [23] proposed a method based on graph neural networks (GNNs) that operates on a KB-subgraph which is enhanced with temporal facts. CʀᴏɴKGQA [46] approaches the problem by obtaining embeddings of answer candidates and temporal values individually. These embeddings are then combined and used for scoring answer candidates directly. TᴇᴍᴘoQR [30] takes a similar approach and augments a question embedding with answer candidate, context, and temporal encodings. Again, this latent encoding is used for predicting an answer score for the embedded answer candidate.

The three methods discussed in more detail represent the state-of-the-art on temporal QA. However, the temporal constraints are handled solely in the latent space, without explicitly (or *faithfully*) pruning out temporally inconsistent answer candidates. Some traditional methods tried to handle implicit temporal questions explicitly, but these approaches were based on handcrafted rules and therefore bound to fail for unseen question patterns (e.g. [22]).

Further, different from general-purpose QA, there has not yet been work on temporal QA that combines heterogeneous information sources for improving the answer coverage.

**Temporal KBs**. More recently, there has been substantial work on temporal KBs or knowledge graphs (KGs) [4, 18, 25, 31, 38, 53, 56]. Such temporal KBs aim to assign a temporal validity to KB-facts. Advances in temporal KBs and their completeness can be seen as orthogonal to this work: enhancing the temporal information covered in KBs would be beneficial for the proposed approach also.

## 6 CONCLUSION

This work proposes a new approach for *faithfully* answering temporal questions, with a focus on the more challenging implicit questions. Experiments indicate a trade-off between faithfulness and answering performance: even if there is no evidence consistent with the temporal constraint, the predicted answer can be correct. Future work could target this trade-off, and identify sophisticated ways to answer faithfully whenever possible, and provide the most relevant answer candidate indicating the *uncertainty* in the explanation for the end user, otherwise.

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

## A  GPT PROMPTS

The prompts ultimately used for creating the training dataset for the generation of intermediate questions can be found in Table 7 and Table 8.

## B  SEQUENCE-TO-SEQUENCE MODELS

There are two different Seq2seq models based on BART in the FAITH pipeline: the model for generating (parts of) the TSF, and the model for generating the intermediate questions. In this section, we would like to give some more technical details on the input and output format of these models.

**TSF construction**. For the TSF construction, the input is only the question, as given by the user. The output is then the concatenation of the individual slots, separated by two pipes ("||"): "{entities} || {relation} || {expected answer type} || {temporal signal} || {temporal categorization}". Example output for $q_1$ (*"Record company of Queen in 1975?"*): "Queen || Record company of in 1975 || record company || overlap || non-implicit".

**Intermediate question generation**. For generating the intermediate question, the input to the BART model is again the question, that has an implicit constraint. The output is then the intermediate question that describes the implicit constraint, and the expected answer type for this question, separated by two pipes ("||"): "{intermediate question}||{expected answer type}". Example output for $q_3$ (*"Queen's record company when recording Bohemian Rhapsody?"*): "when Queen recording Bohemian Rhapsody||time interval".

## C  DISTANT SUPERVISION

The distant supervision follows [12] for obtaining the entity and relation phrase: the entity-centric retriever (see Sec. 2.2) is run on the full question. Recall that this entity-centric retriever identifies entity mentions in the input, disambiguates these to KB-entities, and then retrieves information snippets for the KB-entities. If the retrieved information snippets for a KB-entity contains the gold answer, we treat the corresponding entity mention in the input question as relevant, and add it to the expected output of the training data. The remaining part of the question is kept as relation within the training data. The expected answer type is obtained by looking up all KB-types of the gold answer. The most frequent (proxy for most prominent) of these KB-types is kept as the expected answer type. For dates, years, strings or numbers, we added regular expressions for identifying these.

The temporal signal and the temporal categorization (whether question is implicit or not) is simply looked up from the meta-data available in the benchmarks.

These individual parts are then combined and separated by pipes ("||"), as described in Section B, to obtain the "gold label" outputs for training the TSF construction model (see Sec. 2.1).

## D  QUESTION REPHRASING

The prompt used for rephrasing the pseudo-questions into natural questions can be found in Table 9.

## E  INTERMEDIATE QUESTION DATASET CONSTRUCTION

FAITH requires <questions, date> pairs to train the model for answering the intermediate questions. In TIMEQUESTIONS, there are questions with temporal values as answer that can be used. However, in our new TIQ benchmark, all questions are implicit. We generate intermediate questions using InstructGPT [37] (similar as in section 2.1). The explicit temporal value of the implicit constraint part (from the information snippet in the meta-data of TIQ) is the gold answer to construct <question, date> pairs. If the answer type of an intermediate question is a time interval, we create two questions asking for "start date" and "end date" respectively, as outlined before. We constructed 7,723 questions from the train set and 2,542 questions for the dev set. This augmented dataset will also be made publicly available upon acceptance.

## F  SBERT SCORING MODEL

The set of candidate evidences after temporal pruning can be large (hundreds or thousands), which can affect the efficiency of the answering phase. Therefore, we train a classifier based on SBERT [43] to reduce the size of this set. The training data are the <question, evidence> pairs, annotated with either a positive label (in case they contain the answer) or a negative label (otherwise). We use the concatenation of *question entity*, the *question relation*, and the *expected answer type* of the TSF to represent the question. The pairs are tokenized with the pre-trained language model DistilRoBERTa[5] and fed into the network with their classification labels for fine-tuning the model. Once the classifier is trained, we score each evidence and select the top-100 evidences as input for the answering stage.

## G  ADDITIONAL EXPERIMENTS

### G.1  Temporal signal accuracy

We measure the performance of the temporal signal detection in our TSF construction. In TIMEQUESTIONS, there are 7 classes of temporal signals: *"OVERLAP"*, *"BEFORE"*, *"AFTER"*, *"START"*, *"FINISH"*, *"ORDINAL"* and *"NO SIGNAL"*. In TIQ, there are 3 classes of temporal signals: *"OVERLAP"*, *"BEFORE"*, and *"AFTER"*. We use macro-averaged precision ($\overline{P}$), recall ($\overline{R}$), and F1-score ($\overline{F1}$) as the metrics to evaluate the overall performance of signal detection. The measurements are conducted on the test sets of the two benchmarks respectively. For TIMEQUESTIONS, $\overline{P}$ is 0.898, $\overline{R}$ is 0.888, and $\overline{F1}$ is 0.892. For TIQ, $\overline{P}$ is 0.979, $\overline{R}$ is 0.976, and $\overline{F1}$ is 0.978. The results indicate the feasibility of the proposed signal detection method based on the Seq2seq model. The performance on TIQ is higher than on TIMEQUESTIONS due to the different distribution of temporal signals in the two benchmarks and it is more imbalanced in TIMEQUESTIONS compared to TIQ. Errors when detecting the temporal signal can lead to failures of the temporal pruning stage.

### G.2  Temporal category accuracy

To inspect whether the implicit temporal constraint of a question can be successfully detected, we measure the quality of the temporal category in the TSF. We use precision ($P$), recall ($R$), and F1-score ($F1$) as the metrics. The measurements are conducted on the test set of TIMEQUESTIONS because in TIQ there are only implicit questions. In TIMEQUESTIONS, there are four temporal question categories (explicit, implicit, temporal answer, and ordinal). The questions that

---
[5] https://huggingface.co/distilroberta-base.

do not belong to the implicit category are annotated as the non-implicit type. The performance of the implicit type is as follows. $P$: 0.949, $R$: 0.933, $F1$: 0.941. The performance of the non-implicit type is as follows. $P$: 0.993, $R$: 0.995, $F1$: 0.994. The results indicate the feasibility of the proposed method. An incorrectly predicted category may lead to errors within the implicit question resolver, again resulting in errors during temporal pruning.

## G.3 Implicit question resolver accuracy

The performance of the implicit question resolver is crucial for implicit questions. We use precision ($P$), recall ($R$), and F1-score ($F1$) as the metrics to measure the quality of this stage. We conduct experiments on the test set of TIQ, since we have the ground truth annotated as metadata. The performance is as follows. When $k$=1 (we use the top-1 candidate as the answer to the intermediate questions), $P$, $R$ and $F1$ is 0.532, 0.570 and 0.527, respectively. When $k$=3, $P$, $R$ and $F1$ is 0.342, 0.721 and 0.441, respectively. When $k$=5, $P$, $R$ and $F1$ is 0.224, 0.775 and 0.336, respectively. At $k$=1, $F1$ is the greatest. As $k$ increases, recall improves, but more answers are considered as well leading to noisier evidence that is included. This can negatively affect the system's faithfulness and performance.

**Table 7: Prompt including demonstrations for generating the dataset with intermediate questions on TIMEQUESTIONS.**

Generate an explicit question and answer type for the implicit part of the temporal input question.

Input: what position did djuanda kartawidjaja take after he was replaced by sukarano
Output: when djuanda kartawidjaja replaced by sukarano||date

Input: american naval leader during the world war 2
Output: when world war 2||time interval

Input: who became president after harding died
Output: when harding died||date

Input: who did luis suarez play for before liverpool
Output: when luis suarez play for liverpool||time interval

Input: which countries were located within the soviet union prior to its dissolution
Output: when soviet union dissolution||date

Input: who started the presidency earliest and served as president during wwii in the US
Output: when wwii||time interval

Input: who replaced aldo moro as the minister of foreign affairs
Output: when aldo moro replaced as minister of foreign affairs||date

Input: what did harry s truman work before he was president
Output: when harry s truman president||time interval

**Table 8: Prompt including demonstrations for generating the dataset with intermediate questions on TIQ.**

Generate an explicit question and answer type for the implicit part of the temporal input question.

Input: Who was the second director of the Isabella Stewart Gardner Museum when it was built
Output: When Isabella Stewart Gardner Museum was built||time interval

Input: When Wendy Doniger was president of the Association for Asian Studies, what publishing house was she based in New York
Output: When Wendy Doniger was president of the Association for Asian Studies||time interval

Input: What administrative entity was Ezhou in before Huangzhou District became part of it
Output: When Huangzhou District became part of Ezhou||date

Input: After Bud Yorkin became the producer of NBC's The Tony Martin Show, who was his spouse?
Output: When Bud Yorkin became the producer of NBC's The Tony Martin Show||date

Input: What book did Ira Levin write that was adapted into a film during the same time he wrote the play Deathtrap
Output: When Ira Levin wrote the play Deathtrap||date

Input: What basketball team was Nathaniel Clifton playing for when his career history with the Rens began
Output: When Nathaniel Clifton's career history with the Rens began||time interval

Input: What team did Stevica Ristić play for before joining Shonan Bellmare?
Output: When Stevica Ristić joining Shonan Bellmare||time interval

Input: Which album was released by the Smashing Pumpkins after Mike Byrne joined the band
Output: When Mike Byrne joined Smashing Pumpkins||time interval

**Table 9: Prompt including demonstrations for rephrasing the pseudo-questions into natural questions.**

Please rephrase the following input question into a more natural question.

Input: What album Sting ( musician ) was released, during, Sting award received German Radio Award?
Question: which album was released by Sting when he won the German Radio Award?

Input: What human President of Bolivia was the second and most recent female president, after, president of Bolivia officeholder Evo Morales?
Question: Which female president succeeded Evo Morales in Bolivia?

Input: What lake David Bowie He moved to Switzerland purchasing a chalet in the hills to the north of , during, David Bowie spouse Angela Bowie?
Question: Close to which lake did David Bowie buy a chalet while he was married to Angela Bowie?

Input: What human Robert Motherwell spouse, during, Robert Motherwell He also edited Paalen 's collected essays Form and Sense as the first issue of Problems of Contemporary Art?
Question: Who was Robert Motherwell's wife when he edited Paalen's collected essays Form and Sense?

Input: What historical country Independent State of Croatia the NDH government signed an agreement with which demarcated their borders, during, Independent State of Croatia?
Question: At the time of the Independent State of Croatia, which country signed an agreement with the NDH government to demarcate their borders?

Input: What U-boat flotilla German submarine U-559 part of, before, German submarine U-559 She moved to the 29th U-boat Flotilla?
Question: Which U-boat flotilla did the German submarine U-559 belong to before being transferred to the 29th U-boat Flotilla?

Input: What human UEFA chairperson, during, UEFA chairperson Sandor Barcs?
Question: Who was the UEFA chairperson after Sandor Barcs?

Input: What human Netherlands head of government, during, Netherlands head of state Juliana of the Netherlands?
Question: During Juliana of the Netherlands' time as queen, who was the prime minister in the Netherlands?

