# OpenReview forum: "Faithful Temporal Question Answering over Heterogeneous Sources"
_ACM.org/TheWebConf/2024/Conference — TheWebConf24_

### Official Review · Reviewer_CSX1 · 2023-11-23

**Novelty:** 2
**Technical Quality:** 3

**Review:**

This paper propse a temporal QA system for faithful question answering called FAITH. It consists of three moudules. First, the temporal question understanding module, which transforms the question intent into a structured frame. Second, the faithful evidence retrieval module, responsible for pinpointing pertinent evidence from knowledge bases, text, and tables, incorporating time-sensitive filtering to meet temporal conditions. Third, the explainable heterogeneous answering module, designed to generate entity-level answers and provide supporting evidence for a comprehensive explanation. Additionaly, the authors propose a new benchmark about implicit constraints.

**Questions:**

1. The task format, as well as the input and output of the model, lacks a formal definition. And the paper needs to define what the “faithful” is when mention faithful question answering.
2. The novelty of this work should be further explained. Faithful Evidence Retrieval follow previous entity-centric retriever and Explainable Heterogeneous Answering is based on EXPAINGNN.
3. The method elaboration lacks clarity, resembling a black box where only the input and output are known without knowledge of the intermediate details.
4. The references about the latest temporal question answering are missing. Temporal Reasoning has a lot of literature which could be mentioned.
(1)	A Benchmark for Generalizable and Interpretable Temporal Question Answering Over Knowledge Bases
(2)	Targeted Extraction of Temporal Facts from Textual Resources for Improved Temporal Question Answering over Knowledge Bases
(3)	Towards Benchmarking and Improving the Temporal Reasoning Capability of Large Language Models
5. There are some details:
A. In Section 2.1 Temporal Question Understanding, whether the inferred implicit start date and end date are derived from model or external knowledge. Temporal facts strongly depend on context, the fine-tuned the sequence to sequence model in the closed book setting may be got a bad performance.
B. In Section 2.1 Temporal Question Understanding, the amount of annotated dataset should be make clear.
C. In Section 2.2 Faithful Evidence Retrieval, whether the retrieval strategy is different under different data sources.
D. In Section 2.2 Faithful Evidence Retrieval, the granularity of the evidence is not discussed. If each evidence is a sentence , your temporal pruning policy will lose information when multiple sentences share the same time.
E. In Line 569, more train details should be shown such as batch size, learning rate, optimizer, epochs and so on.
F. In Line 613, the author retains 100 evidences into models, whether the amount of input evidence influence the performance of the model.
G. In Table 3, more reason about why UnFAITH is better than FAITH should be talked. And it appears to introduce a certain imbalance: GPT receives input solely in the form of a question, whereas other models are provided with both the question and the retrieved evidence. What will be when GPT receieve question and evidence.

**Reviewer Confidence:**

4: The reviewer is certain that the evaluation is correct and very familiar with the relevant literature

**Scope:**

3: The work is somewhat relevant to the Web and to the track, and is of narrow interest to a sub-community

---

### Official Review · Reviewer_UkPr · 2023-11-24

**Novelty:** 4
**Technical Quality:** 6

**Review:**

The paper proposes FAITH (FAIthful Temporal question answering over Heterogeneous sources) - A temporal Question Answering (QA) model that answers implicit temporal questions using text, KG, and infoboxes. To make the answer faithful,  the QA model also provides the actual source where the answer is drawn from and uses a GNN-based explainable answer retriever component. Besides that, the work introduces a Temporal QA dataset (size: 10,000) to evaluate their model.

# Strengths
* The paper is well-written.
* Adopt the methods used in the generic heterogeneous QA to Temporal QA.
* Broden the scope of Temporal QA from one source to multiple heterogeneous sources, namely KG, text, and infoboxes.
* Create a Temporal QA model and a benchmarking dataset.
* Provide evidence along with answers to increase the faithfulness of the answers.

# Weaknesses
* Usage of entity-centric retriever.
* In section 3, It is stated that the existing TQA benchmarks ‘have only a few implicit questions’. To know how small they are, this paper does not have any tabular data to compare the number of implicit questions in the listed existing dataset.
* Given that a total ~ 43% = KB; KB + Infobox; Infobox + Text; Text questions in the TIQ dataset are answerable using one source, would be nice to put the performance of the FAITH model on those real heterogeneous questions only in the dataset.
* Even though the questions are generated and rephrased using an LLM, this method doesn’t assure the questions’ syntactic diversity. In addition, the questions are generated using a limited number of examples (only ‘8 pseudo-questions with their natural rephrasing’), this makes the diversity of the questions in TIQ limited.
* The paper does not provide a statistical distribution of the question types (What, who, where, …) in the TIQ dataset.

**Questions:**

* It is obvious using heterogeneous data sources makes answer retrieval very challenging. However, irrespective of the data source the question understanding part of your model, how is it different from other heterogeneous QA models?
* Does only providing the evidence for the answer to a question make the model faithful? What about the process to reach the answer?
* All the retrieved pieces of evidence are verbalized, so how do you use text data to a GNN model, at the explainable answering stage?
* While retrieving evidences do you follow the same method for retrieving from the different sources? How does the heterogeneous retriever deal with the differences among the data sources?
* As shown in Figure 3, there is no question that needs Table source reasoning, why do you need Table, KB + Tables, Text + Tables, and Tables + Infoboxes evaluation in Table 4?
* In Section 5 (lines 847 - 849), it says “However, LLMs cannot present traceable provenance for the provided outputs, and faithfulness and explainability are key concerns [1, 29, 32].”; Saying that how do conclude that your model is faithful being tested on an LLM generated dataset?

**Ethics Review Description:**

-

**Reviewer Confidence:**

3: The reviewer is confident but not certain that the evaluation is correct

**Scope:**

4: The work is relevant to the Web and to the track, and is of broad interest to the community

---

### Official Review · Reviewer_PrGV · 2023-11-24

**Novelty:** 4
**Technical Quality:** 5

**Review:**

The authors introduce a new approach to temporal question answering, which involves the understanding, retrieving, and processing of temporal information. The authors are particularly interested in implicit questions, where temporal values are not directly mentioned in the questions but via temporal relations with other facts. The proposed model can retrieve background information from heterogeneous sources including text and tables from Wikipedia and the knowledge base Wikidata. The authors also introduce a new dataset TIQ with rich implicit questions, and evaluate the proposed model on both existing and the new datasets.

The proposed approach appears to be based on ExplaiGNN [12] for background information retrieval and answer generation, with additional modules for resolving implicit questions and temporal pruning of generated answers. Hence, it would be useful to discuss the novel technical contributions in comparison with ExplaiGNN. In particular, it is interesting to see a discussion on why the variant Un-Faith not utilizing the temporal constraints can outperform ExplaiGNN.

An interesting observation, which could be a major limitation of the paper, is that Un-Faith outperforms the proposed model Faith on both existing and new datasets. The authors explain this as Faith refraining from answering in the absence of evidence consistent with the temporal constraints. I wonder if it makes sense to consider "no answer" as the correct answer in such a case. In the error analysis, it shows over 80% and 60% of the failure cases on respectively TIQ and TimeQuestions are due to temporal pruning. I wonder if this reflects issues in the datasets themselves, especially for the new dataset, and whether it is possible to construct it in a way that models do not have to choose between faithfulness and answering performance.

Moreover, some technical details could be further clarified. The temporal signal can be overlap, before, or after, while other common temporal relations could be considered, such as contain/cover and within. Also, for the fine-tuning of BART for resolving implicit questions. It is said that 8 implicit questions are selected and labelled, and then it is said a large-scale data is obtained for fine-tuning. It is unclear to me how the large-scale data is generated from the initial 8 questions. Another question is related to how the temporal pruning is done and whether this is an accurate process.

Finally, some experiment designs could be further explained. Why TimeQuestions is adopted but not TempQuestions, TempQA-WD, or CronQuestions. Also, why only the vanilla version of ExplaiGNN is used as a baseline?

**Questions:**

Q1: What are the novel technical contributions compared to ExplaiGNN, and why does Un-Faith outperform ExplaiGNN?

Q2: Would it be possible to refine TIQ by reducing failure cases due to unfaithful answers? Should not these unfaithful answers be considered incorrect?

Q3: Why TimeQuestions is adopted but not TempQuestions, TempQA-WD, or CronQuestions, and why only the vanilla version of ExplaiGNN is used as a baseline?

**Reviewer Confidence:**

3: The reviewer is confident but not certain that the evaluation is correct

**Scope:**

3: The work is somewhat relevant to the Web and to the track, and is of narrow interest to a sub-community

---

### Official Review · Reviewer_T5tY · 2023-11-25

**Novelty:** 4
**Technical Quality:** 4

**Review:**

The work aims to approach temporal question answering with faithfulness. The method has three stages: (i) understanding the question and its temporal conditions, (ii) retrieving evidence from all sources, consistent with the temporal constraints, and (iii) faithfully answering the question from these pieces of evidence. As implicit questions are largely underrepresented in established benchmarks, authors introduce a principled method for generating diverse questions of this kind in a systematically controllable way from heterogeneous sources.

Overall, the experiment results are good, and the authors put a visible effort in explaining the paper well.

Let me now summarize key points I noticed and appreciate if authors take time during rebuttal to answer them. I promise I will read your comments and acknowledge them after rebuttal. And then we wait until post-Christmas to see the final result of the paper (Good luck :)).
Let us come back to paper :)->

  - What is faithfulness in scope of paper? I miss a formal definition of this term. How does it relate to faithfulness metric in NLP community? Can authors be bit more formal in justifying how can proposed method be grounded on formal definition/metric of faithfulness? The proposed approach seems quite  a black-box and I am unable to connect its various bits and pieces to make a clear end-to-end picture.
  - When it comes to explainability, there is also a lot of work in explaining HCI systems. For instance, work in {1,2} put several formal parameters to evaluate explainability of the system. Can authors comment why such metric has been omitted from the empirical result inspite clear focus on explainability? Questions such as 1)“How effective is our approach for generating explanations? 2) How effective is the perception of the end-user?
  - Coming back to end-user, I do miss human-evaluation in the experiment section. Although I like the idea of faithfulness in the work, I clearly miss formal grounding of definitions, then connecting them to evaluation section and human evaluation to quantitatively judge “scope-of-faithfulness”.

1. Upol Ehsan, Pradyumna Tambwekar, Larry Chan, Brent Harrison, and Mark O. Riedl. 2019. Automated rationale generation: a technique for explainable AI and its effects on human perceptions. In Proceedings of the 24th International Conference on Intelligent User Interfaces, IUI 2019, Marina del Ray, CA, USA, March 17-20, 2019, pages 263–274.

2. Upol Ehsan and ark Riedl. 2019. On design and evaluation of human-centered explainable ai system. In Human-Centered Machine Learning Perspectives Workshop at CHI.

**Questions:**

Please see above.

**Reviewer Confidence:**

4: The reviewer is certain that the evaluation is correct and very familiar with the relevant literature

**Scope:**

4: The work is relevant to the Web and to the track, and is of broad interest to the community

---

### Decision · Program_Chairs · 2024-01-22

**Decision:**

Accept

**Comment:**

This paper introduces a novel approach to temporal question answering, extending the scope to multiple heterogeneous sources and presenting the TIQ dataset for implicit questions with temporal relations. The proposed model aims for faithfulness, demonstrated through experiments and benchmarking.
 Advantages include good experimental results, broadened Temporal QA scope, model creation, and evidence provision for enhanced faithfulness.
 The authors response effectively addressed some of the weaknesses and concerns from the reviewers, including the absence of human evaluation, limited question syntactic diversity, missing question type distribution in the TIQ dataset, and Un-Faith outperforming Faith. The authors address these in the rebuttal, clarifying faithfulness and technical contributions, promising additional analysis and references in the camera-ready version.
 Considering the targeted complexity of temporal questions and specific contributions, overall, the paper makes a valuable contribution with potential enhancements in experimental design and clarity.
 To the authors please make sure all your answers and clarifications to reviewers questions are included in the camera ready